# Solution pH Effect on Drain-Gate Characteristics of SOI FET Biosensor

**Anastasia Bulgakova** [1], **Anton Berdyugin** [1], **Olga Naumova** [2], **Boris Fomin** [2], **Dmitrii Pyshnyi** [1], **Alexey Chubarov** [1,*], **Elena Dmitrienko** [1] **and Alexander Lomzov** [1,*]

1   Institute of Chemical Biology and Fundamental Medicine SB RAS, 630090 Novosibirsk, Russia
2   Rzhanov Institute of Semiconductor Physics SB RAS, 630090 Novosibirsk, Russia
*   Correspondence: chubarovalesha@mail.ru (A.C.); lomzov@niboch.nsc.ru (A.L.);
    Tel.: +7-913-763-1420 (A.C.); +7-913-009-2889 (A.L.)

**Abstract:** Nanowire or nanobelt sensors based on silicon-on-insulator field-effect transistors (SOI-FETs) are one of the leading directions of label-free biosensors. An essential issue in this device construction type is obtaining reproducible results from electrochemical measurements. It is affected by many factors, including the measuring solution and the design parameters of the sensor. The biosensor surface should be charged minimally for the highest sensitivity and maximum effect from interaction with other charged molecules. Therefore, the pH value should be chosen so that the surface has a minimum charge. Here, we studied the SOI-FET sensor containing 12 nanobelt elements concatenated on a single substrate. Two types of sensing elements of similar design and different widths (0.2 or 3 μm) were located in the chips. The drain-gate measurements of wires with a width of 3 μm are sufficiently reproducible for the entire chip to obtain measurement statistics in air and deionized water. For the pH values from 3 to 12, we found significant changes in source-drain characteristics of nanobelts, which reach the plateau at pH values of 7 and higher. High pH sensitivity (ca. 1500 and 970 mV/pH) was observed in sensors of 3 μm and 0.2 μm in width in the range of pH values from 3 to 7. We found a higher "on" current to "off" current ratio for wide wires. At all studied pH values, $I_{on}/I_{off}$ was up to 4600 and 30,800 for 0.2 and 3 μm wires, respectively. In the scheme on the source-drain current measurements at fixed gate voltages, the highest sensitivity to the pH changes reaches a gate voltage of 13 and 19 V for 0.2 μm and 3 μm sensors, respectively. In summary, the most suitable is 3 μm nanobelt sensing elements for the reliable analysis of biomolecules and measurements at pH over 7.

**Keywords:** silicon-on-insulator; field-effect transistors; SOI-FET; silicon nanobelt; nanowire; biosensor; drain-gate characteristics

## 1. Introduction

Detection of nucleic acids (NA), proteins, and other biological targets are essential in clinical diagnostics, modern molecular biology, food safety, and environmental surveillance [1–6]. The application of label-free biosensors based on silicon-on-insulator (SOI) field effect transistors makes detection fast, simple, and easy [7]. The high signal-to-noise ratio (SNR) for SOI sensors and low detection limit reaching femtomolar concentrations of the biomarker [8–11] make their usage very attractive. These two aspects lead to several problems in this type of biosensor development. Many factors affecting sensor properties should be considered. The sensor element type and design [4,9,12–14], detection environments (in air or solution), solution properties and characteristics of an analyte, and surface modification should be taken into account in the design of the biosensor. All these factors are interrelated and impose utter limitations on the design of the biosensors.

A reference should be used to provide reliable and reproducible analysis. In the case of SOI-FET sensors, multiple microwires as part of a single chip is a convenient choice. The

detection of the biomarkers based on the specific interaction with the ligand is performed in solution, with solution parameters such as ion concentration and type, pH value, and the presence of co-solvent effects on the target detection [8,15–19]. It is essential both for sensing element and probe-target interactions.

The ion concentration in the buffer solution plays an important role in the detection efficiency of analytes. A high ion concentration can significantly decrease the sensitivity of FET biosensors [20]. The electrostatic screening by the ions in the electrolyte near the surface of the sensor element is one of the key factors affecting the sensitivity (changes in the conductivity of the sensing element) of FET sensors. However, electrostatic screening depends on the sensor element design. For example, it is stronger near the convex/concave sensor surfaces than near the flat surface of the rectangular sensor (edge effect) [21]. The conditions under which the probe-target complex will be stable should also be considered when designing a biosensor. The ionic strength is crucial for biopolymer interaction, especially for polyanion chains of nucleic acids. For most cases, high concentrations of cations are required for specific complementary complex formation. There are just a few exceptions based on the usage of uncharged nucleic acids derivatives, such as a phosphoryl guanidine oligonucleotide, as an acceptor on the sensor surface, allowing the detection of nucleic acids in pure water [22]. Surface chemical modification of the sensing surface is required to attach a specific probe to the sensor. The type of surface modifier depends on the immobilized molecules [23]. The modification can significantly affect the biosensor response and the ability to detect the analyte [22].

The pH value of a solution is also essential for detection. Biopolymers could degrade at low or high pH values. For example, RNA hydrolyzes in alkaline pH, and DNA molecules are in an acidic environment. The spatial structure of proteins is destroyed, which affects their specific interactions. The effects of solution pH can be significant and depend on the SOI-FET sensor element size [8] and measurement scheme [24]. Analysis of the widths of channels of 130, 150, and 220 nm showed the channel conductivity increased as the acidity of the buffer solution decreased. At higher pH levels, the current in the channel first increased sharply and then went into the saturation region. The current increased slightly over the measurement time in the saturation region. Stable and reproducible results were found for biosensors of 220 nm width. Obviously, the smaller the sensor width, the higher the contribution of the edge effect to the sensor conductivity.

The pH value affects the surface charge and, thus, the characteristics of the sensors. The silicon surface of the biosensor is covered by a thin layer of silicon oxide. After surface oxidation, the silicon hydroxide is formed on its surface. Protonation of silanol groups occurs in an acidic environment, and their deprotonation is under an alkaline environment (Figure 1C) since the pKa for silanol groups (SiOH) on the surface is 4.5 to 8.5 [25,26]. However, S. Zafar et al. report that the pH sensitivity of nanowires is independent of the buffer solution concentration and that the mechanism of nanowires sensing in ionic solution is complicated [27]. The pH values may change during the biochemical analysis [28]. Consequently, the analysis conditions, including pH values, have a significant impact on both the state of the sensor and the efficiency of its interaction with the molecule. It is important to select the pH value or range in which this influence will be minimal. The measurement scheme depends on the pH value [24]. In the pH sensitivity of sensors, as a rule, an electrode placed in an electrolyte (reference electrode, RE, Figure 1B) is used as a control gate [8,27]. To prevent the charge of biomolecule changes at the detection, the conductivity is typically controlled by the voltage on the SOI substrate (i.e., back-gate (BG) SOI-FET sensor). In this case, the RE is grounded [24,29–34]. The bottom gate scheme showed a very high sensitivity of the silicon nanowire-based pH sensor, reaching 1.4 V/pH [24].

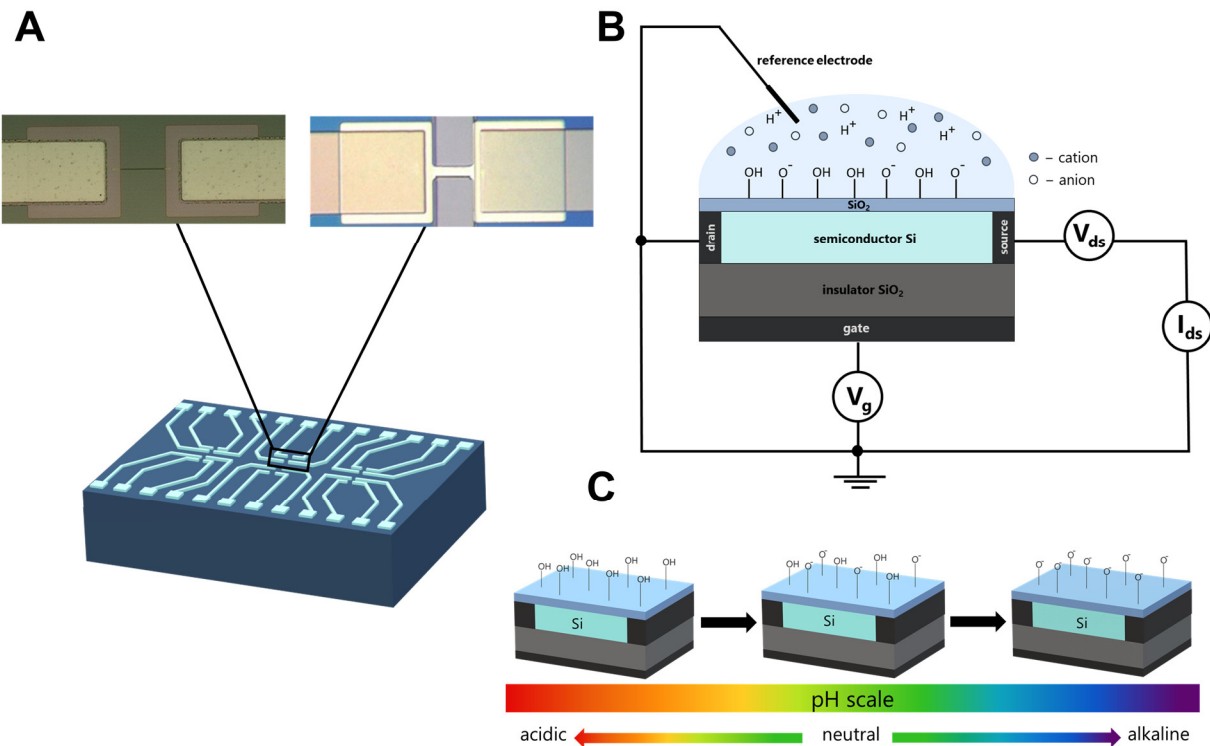

**Figure 1.** (**A**) Sensing element structure with 12 individual transistors, nanobelt scheme, and optical microscopy images of nanobelts of 0.2 and 3 μm in width. (**B**) The measurement scheme for every transistor's channel. (**C**) Schematic representation of pH value effect on the charge state of the SOI-FET surface.

To consider the above effects, a reference sensing element can be used, which must be subjected to the same influences as the working one. It can be placed on the same substrate and used for lab-on-chip construction. Our previous studies showed the high sensitivity of SOI FET sensors for biomolecule detection in model laboratory conditions [8]. The real samples will have various impurities that can eliminate the profits of the sensing elements. The various factors should be assessed separately to analyze the possibility of the usage of SOI FET in the bioanalysis.

In this study, we investigate the effect of pH value on the SOI-FET response. The n-channel back-gate (BG) SOI-FET sensors with silicon oxide surfaces of 3 μm and 0.2 μm width were used. We analyzed the reproducibility of the measurements for different nanobelts on the single chip to select the nanobelt parameters for the most reproducible measurements. The drain-gate characteristics were analyzed in air and solutions at different pHs varied from 3 to 12. We used clean, unmodified microwires unified to avoid unrelated side effects. It was found that the 3 μm sensors give a more reproducible result. The pH sensitivity and "on" current to "off" current ratio for the wider sensor are reliably higher. The threshold voltage for both types of sensors rises at a pH increase and reaches the plateau at a pH higher than 6. The data obtained indicate that the 3 μm width sensors are more suitable for biosensor applications, and pH values at the analyte detection should be 7 or higher for reliable and reproducible analysis.

## 2. Materials and Methods

### 2.1. Materials

Sodium acetate (99.99%), acetic acid (100%), sodium hydroxide (99.9%), potassium dichromate (99.5%), and sulfuric acid (0.5 M) were purchased from Sigma (St. Louis, MO, USA) at the highest available grade and used without purification. Deionized water (Milli-Q) was used to prepare solutions. The following solutions were prepared: 200 mM sodium

acetate buffer, which was adjusted to the pH 3.0–6.0 by acetic acid; 200 mM sodium acetate solution, which was adjusted to the pH 7.0–12.0 by sodium hydroxide.

### 2.2. SOI-FET

We used n-channel back-gate (BG) SOI-FET sensors manufactured by a top-down fabrication process based. Figure 1A shows the schematic representation sensor containing 12 SOI-FET and photography of sensing nanobelts of different width. The chips with 3 µm width sensors were manufactured based on Rzhanov Institute of Semiconductor Physics of the Siberian Branch of the Russian Academy of Sciences by lithography. The main steps of manufacturing sensors are described elsewhere [35]. The chips with 0.2 µm width sensors were provided by MBU (the Medical-Biological Union) Technology LLC (Moscow, Russia). The length of the sensor element was 10 µm (top surface area 30 and 2 $\mu m^2$ for the first and the second cases, respectively). The sensors have the 30*30 $\mu m^2$ source-drain gate pads. Starting SOI wafers have 30 nm thickness of the top Si layer, and 200 nm thickness of the buried oxide. The constructional parameters of both types of chips were the same, except for the width of sensing elements.

### 2.3. Surface Preparation of SOI-FET Sensors

Silicon sensors were cleaned in potassium dichromate solution in sulfuric acid at 25 °C for 15 min, followed by rinsing in copious deionized water, to clean the surface and generate surface silanol (Si–OH) groups. Then, sensors were washed with ethanol, and acetone, dried under air, and used for analysis.

### 2.4. Measurement Technique

Typically, when measuring the sensor pH sensitivity, a buffer solution is used as the gate, and the gate voltage is applied to the electrode immersed in the solution $V_{RE}$ [35] (see Figure 1A). The sensing signal is the drain-source current ($I_{ds}$), which is measured as a function of gate voltage ($V_{RE}$) whilst the drain-source voltage ($V_{ds}$) and SOI-substrate voltage (Vg) are held constant. In our case, the SOI substrate was used as the back-gate (BG) to control the sensor conductivity. The voltage on the electrode immersed in the solution was equal to zero. A Keithley 6487 picoamperemeter/voltage source (Keithley Instruments, Solon, OH, USA) was used to generate the target voltages on the SOI substrate used as a gate of sensors. A constant voltage of 0.19 V was applied between the source and the drain, then the voltage was measured, produced by the operational amplifier included in the measurement block. Electrical measurements were made using an Agilent 34410A multimeter (Keysight Technologies, Santa Rosa, CA, USA). The drain-gate characteristics (dependence of source current on gate voltage) of each sensor were measured with a gate voltage sweep from 0 to 30 V in steps of 1 V. The circuit used a bottom-gate implementation. The ratio method was used to determine the value of the threshold voltage from the measured drain current versus gate voltage transfer characteristics [35,36].

### 2.5. Device Calibration

To obtain the drain-gate characteristics in the mode of volt-ampere dependences, the measuring unit was calibrated using a set of resistors with different resistances, which were measured beforehand using an Agilent 34410A multimeter.

### 2.6. Investigation of pH Effect on the Sensor Response

The effect of pH on the sensor response was studied using chips with a clean, un-modified surface of nanobelts. The 20 µL of 200 mM sodium acetate solutions with pH values from 3 to 12 was placed on the surface of the sensor. The dependence of source current on gate voltage was collected through consecutive measurements of 12 nanobelts at every pH value.

## 3. Results

### 3.1. Calibration of Source Current Values

We developed a device that can successively measure the drain-gate characteristics of 12 nanobelts on a single chip [37] (Figure 1). For high-accuracy current measurements, we used an operational amplifier as a part of the device. We applied a set of resistors to calibrate the setup to convert the voltage value measured by the voltmeter into a current value. The resistance was measured in a two-wire scheme using an Agilent 34410A multimeter. Analysis of the estimated current values shows the relative error in the current determination was less than 1.5% over the entire measurement range ($0.01-7$ μA). At the same time, the reproducibility of repeated measurements was an order of magnitude lower, at the level of 0.1%. It makes it possible to use the obtained calibration line for calculating the drain-source characteristics in the mode of dependence of the source current on the voltage on the substrate. Moreover, automating and standardizing the measurement process, as well as facilitating the processing of the obtained data, can be obtained.

### 3.2. Reproducibility of the Measurements

We analyzed the reproducibility in a series of measurements of the drain-gate characteristics of silicon microwires with a width of 0.2 and 3 μm with a clean surface. Measurements were performed in air and in deionized (Milli-Q) water (resistance 18.2 MOhm) at the chip surface using the scheme presented in Figure 1B. All 12 wires 3 μm wide on the chip have very similar trends in air and water. In air, the current values are slightly shifted relative to each other but the general form of the dependences repeats each other well. Repetitive measurement of the same wire does not exceed 6% in the whole voltage range in all experiments in the study.

The average value of the threshold voltage measured in air was $8.66 \pm 0.76$ V. When the sensor elements are placed in the water, the current values for the different wires almost completely repeat each other in the whole gate voltage range (Figure 2A, blue lines). The maximum value of RMSD in a series of measurements in air and water differs 2.7 times. The average value of the threshold voltage measured in water was $20.7 \pm 0.28$ V. In general, the repeatability of measurements in water for different wires is better than in air, and they have the same type of volt-ampere characteristics.

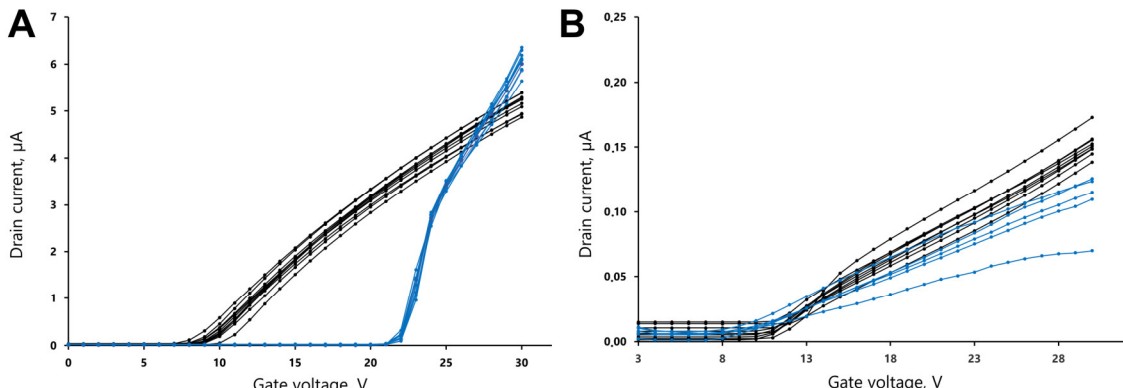

**Figure 2.** The drain-gate characteristics of the transistors in air (black) and water (blue). (**A**) 3 μm wires; (**B**) 0.2 μm wires. The error of the point measurement does not exceed 6% at a gate voltage higher than a threshold voltage.

For 0.2 μm wide wires, sufficiently reproducible drain-gate characteristics are registered in air. The average value of the threshold voltage measured in air was $9.56 \pm 0.72$ V. For wires on the same chip, measurements in water vary significantly. However, only 5 of 10 wires on one chip keep normal types of volt-ampere characteristics with an average threshold voltage value of $8.50 \pm 0.52$ V. The remaining five wires lose their normal current-voltage characteristics, moving to negative current values. Similar trends were observed

for four different chips containing wires with a diameter of 0.2 μm. It is probably due to the leakage current. We can conclude that the drain-gate measurements of wires with a width of 3 μm are sufficiently reproducible for the entire chip to obtain measurement statistics in air and deionized water. For wires with a width of 0.2 μm, sufficient correlation of results between different sensors of the same chip was observed for measurements in air.

### 3.3. pH Effect on the Drain-Gate Characteristics

We analyzed the effect of pH value on the drain-gate characteristics. The buffers with 200 mM sodium acetate concentration and pH values in the range of 3–12 were selected. The high ionic strength was chosen to prevent the effects of the changes in the ionic strength at different pH values. The increase in pH leads to the shift of the drain-gate characteristics to the higher voltage range (Figure 3), as described previously [24,29,30,33,34,38–40]. For some 200 nm width nanobelts in the same chips at high pH values, the drain gate does not change reproducibly. For a clear presentation of the results obtained and a more reliable analysis, individual wires with the most repeatable drain-gate characteristics were used.

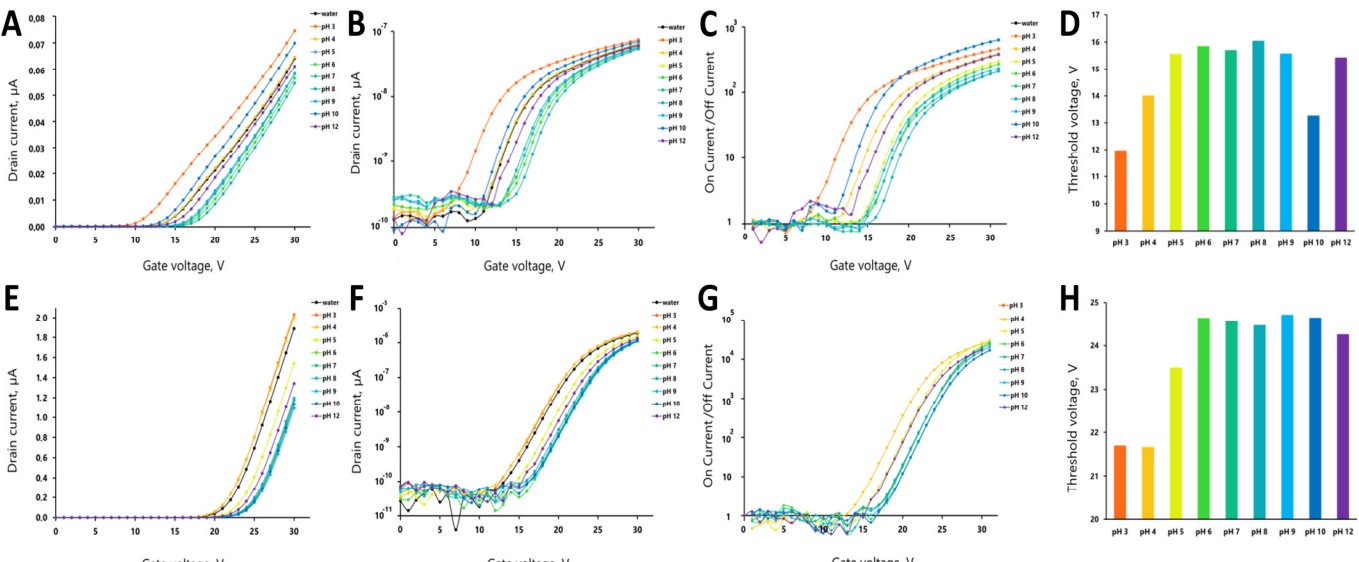

**Figure 3.** The drain-gate characteristics analysis of the sensors as a function of solution pH (200 mM sodium acetate). (**A–D**) characteristics for 0.2 μm SOI-transistor; (**E–H**) for 3 μm SOI-transistor. (**A**,**E**,**B**,**F**) are drain-gate characteristics of a single wire over the entire range of solution pH values in linear and logarithmic scales. (**C**,**G**) "On" current to "off" current ratio curves. (**D**,**H**) The dependence of threshold voltage on the pH.

We performed the analysis of the obtained drain-source characteristics. A better "On" current to "Off" current ratio for wide wires was found. At all studied pH values, $I_{on}/I_{off}$ were up to 4600 and 30,800, decreasing ca. two times at pH increase for 0.2 and 3 μm wires, respectively.

The threshold transition voltage was calculated using the ratio method [36]. An increase in the threshold transition voltage from 21.7 to 24.5 V with an increase in pH values from 3 to 12 was observed for 3 μm wires (Figure 3D). The maximum transition was observed between pH 4 and 6. The sensitivity of the wide sensor in this pH region was 1.49 V/pH. In contrast, for the 0.2 μm sensors, an increase in the threshold transition voltage from 12.0 to 15.8 V was observed with an increase in pH values from 3 to 12 (Figure 3).

The main changes were in the range of pH from 3 to 6. The sensitivity of the narrow sensor in this pH region was 0.97 V/pH. Some perturbations of the measurements at the pH range of 10 and 12 originated from the low buffer capacity at this range. In some cases, the biosensor's measurement was performed at a fixed gain voltage by registration of the source-drain current at the time of adding a target for detection. The analysis of the

current at the gate voltages higher than the threshold voltage on the pH showed the inverse dependence presented in Figure 3D,H.

We analyzed the source-drain current at fixed gate voltages. We selected the lowest pH value 3, where the silanol surface is fully protonated, and pH 7 as a value, where the pH dependence reached a plateau and the surface became deprotonated. The dependence of the source-drain current's relative changes at pH 3 and 7 (($I_{ds}$(pH 3)- $I_{ds}$(pH 7))/$I_{ds}$(pH 7)) on gate voltage was obtained for both nanobelt types (Figure 4).

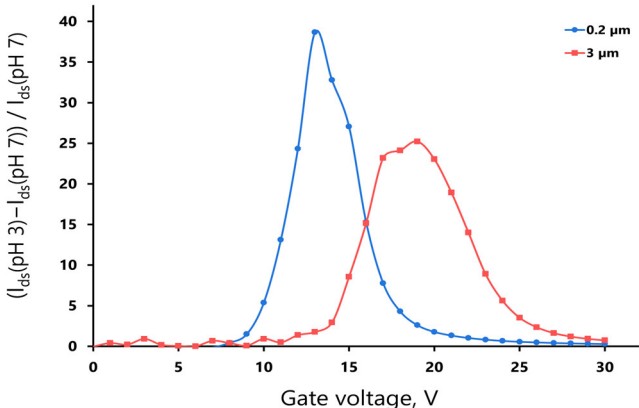

**Figure 4.** Dependence of source-drain current relative changes at pH 3 and 7 on gate voltage for 0.2 μm (blue) and 3 μm (red) SOI-transistor.

This dependence shows the way to select the optimal gain voltage for the measurements of wire conductivity by varying any factor affecting it. In our case, it was changed in pH value. The values of the current changes for different pH values strongly depend on the Vg selected, reaching the relative change values of 35 and 29 at gate voltage 13 and 19 V for 0.2 μm and 3 μm sensors, respectively. For our sensors, such voltages corresponded to the highest sensitivity to the pH changes.

## 4. Discussion

The SOI-FET transistors for biosensing applications are very attractive due to the possibility of fast, simple, and easy label-free detection of bioanalytes. Previously, it was shown that these sensors make it possible to detect the very low quantity and/or concentration of the probe. This high-sensitivity analysis is typically performed in the "ideal" conditions with only probe and buffer presented in the analyzing solution. However, a number of problems are limiting its application in medical diagnostics. One of the issues is the significant effect of non-specific components presented in the solution on the signal of the sensor. The influence of these factors should be taken into account; mainly, buffer conditions (ionic strength and pH) and the presence of non-target components which cannot be removed from the analyzing solution.

The performance of biosensors based on field-effect transistors depends on the environment in which measurements are carried out due to the screening effect. In an aqueous solution, the surface charge attracts counterions forming an electrical double layer that screens the surface charge [20,41]. The physical distance of this screening action is characterized by the Debye length, which under physiological conditions, is usually about 10 nm [42]. As the ionic strength of the solution increases, the electrical double layer and Debye length decrease because of charge screening by counterions. As a result, the sensitivity of the device is reduced. Precisely, the changes in charge density that occur within the Debye length of the sensor surface can significantly affect the threshold voltage of the transistor [43]. The Debye length corresponds to the distance from the charged surface, at which the surface potential drops by a factor of 2.7 compared to the initial value due to charge screening by counterions. The screening is modeled as an exponential decay

function. The Debye length depends on the ionic strength of the solution and decreases as the ionic strength of the solution increases, as can be seen from the equation:

$$\lambda_D = \left( \varepsilon k_B \times \frac{T}{q^2 I} \right),$$

where $\lambda_D$ is the Debye length, $\varepsilon$ is the dielectric permittivity of the medium, $k_B$ is the Boltzmann constant, $T$ is the absolute temperature, $q$ is the electric charge, and $I$ is the ionic strength of the solution.

Because of the dependence of the Debye length on the ionic strength of the solution, field-effect transistors show low sensitivity in biological samples such as serum. Because of this, most studies are performed under low ionic strength conditions (10 mM) [44].

The effect of pH was studied for various sensors (Table 1) [24,29,30,33,34,37–39]. The effect of SOI sensors' parameters (sensor element design, size, materials, coverage, etc.), measurement scheme, and different buffers was analyzed. In most cases, the pH-driven changes in sensors were studied in the pH sensors' development. In contrast, likewise, an effect does not take into account in biosensor design. Nonetheless, the pH-induced effect can be significant in biological probes analysis. It can affect probe–target interaction. On the other hand, the changes in the sensor surface charge originating from pH changes can significantly influence selectivity, sensitivity, and the possibility of probe detection. The probe molecules covalently bind to the sensor's surface for selective target detection; in this case, the nanobelt surface modifying with various chemical compounds. Despite these, the silanol groups can still be presented on the surface as functional groups. It is necessary to analyze the pH effect on the clean silanol surface of the nanobelt to exclude other effects.

**Table 1.** The effect of pH for various sensor systems.

| Nanowire Materials | Gate Type | Buffer Solutions | pH Range | pH Sensitivity | Ref. |
|---|---|---|---|---|---|
| Hydrogel-gated Si nanowire | Top gate | 0.01 M phosphate buffer | 3, 5, 7, 9, 11, 13 | 100 mV/pH | [34] |
| Silicon nanowire | Top gate and back gate | Buffer solution | 6.3, 7, 8.2 | 59.5 mV/pH | [33] |
| Silicon nanowire | Top gate and back gate | Various buffer solutions | 3.07 to 9.87. | 56.3 mV/pH (single gate) znj1438.8 mV/pH (dual gate) | [24] |
| Silicon nanobelt | Back gate | 0.1 x–10 x phosphate-buffered saline | 4 to 8 | 57.2 mV/pH. | [32] |
| Poly-Si nanowire | Top gate and back gate | Different pH buffers | 3, 5, 9, 10, 11 | 178 mV/pH | [37] |
| TiO$_2$—polyaniline composite thin films | Back gate | pH buffer solutions | 2, 4, 7, 10, 12 | 66.1 mV/pH | [38] |
| Solution-gated reduced graphene oxide | Top gate | 0.1 M phosphate buffer | 6 to 9 | 29 mV/pH | [30] |
| Aluminum oxide (Al$_2$O$_3$)-gate | Top gate | pH buffer solutions (Merck) | 4, 7, 10 | 42.1 mV/pH | [29] |
| In$_2$O$_3$ nanobelt | Back gate | Standard commercial pH solutions | 6 to 10 | 88.125 mV/pH | [39] |

The reference sensing element without specific binding of the target is necessary for molecular diagnostics systems development. It can be realized in the lab-on-chip design using, for example, a single substrate for several sensing elements to minimize side effects. For this purpose, the obtaining signal should be reproducible and stable at the same time

for different sensing elements. Here, we analyzed the pH effect on the two sizes of nanobelt SOI-FET sensors that showed high perspectives for nucleic acid sensing, which we studied previously [8]. We analyzed the pH dependence of 12 nanobelt SOI-FET sensor elements of different widths (0.2 and 3 μm) concatenated on the single substrate (Figure 2). We found higher reproducibility for different wires of the source-gain characteristics for wider nanobelt measured in air and deionized water. However, we have shown the similarity of response sensors located on the same substrate highly decreases for narrow nanobelts, which affects the results' reliability. The reproducibility in water was about 50% for 0.2 μm wires. Therefore, analysis in an aqueous medium on 3 μm wide nanobelt transistors is preferable. Measurements in water give a similar current-voltage characteristic to wide wires with close to 100% repeatability. The state of the SOI-transistor surface before and after measurements in solution also affects the drain-gate characteristics.

We have analyzed the pH range of 3–12 to cover the range of pKa values for silanol group presented in the literature (from 4.5 to 8.5) [25]. Therefore, at a pH lower and higher than the studied one, the surface of our sensors should be fully deprotonated and protonated, respectively. Analysis of source-drain characteristics showed the shift of the threshold voltage to the higher values. It is consistent with the previously published data for SOI FET pH sensors [24,29,30,33,34,37–39].

We found the most significant changes in the source-drain characteristics in the range up to pH 6 (Figure 3). At a pH value of 7, the dependence reaches the plateau, which originated by full deprotonation of the sensor surface. A significant shift of threshold voltage resulting in high pH sensitivity (ca. 1500 and 970 mV/pH) was observed at 3 μm and 0.2 μm width sensors in the studied range of the pH values. The main origin of the effect observed is in the surface protonation of the sensor during interaction with a solution. Moreover, the sensor of various widths will have a different edge-effect that takes a contribution to the observed changes in the drain-gate characteristics [27]. These results indicate that the optimal pH value is in the plateau region, starting at the value of 7. At the lower pH values, the source-drain characteristics can change significantly at small changes in pH value.

The data demonstrated in Figure 4 shows the way to select optimal gain voltage for the measurements of wire conductivity at target detection in biosensors. The reference sensing element is obviously necessary for molecular diagnostics systems development. The analysis of the source-drain current ratio on gate voltage at probe analysis should be performed for reference and targeted sensing elements. Based on their comparison, the gain voltage for this measurement scheme can be performed to reach the maximum difference between them.

## 5. Conclusions

Herein, we studied the SOI-FET sensor containing 12 nanobelt elements concatenated on a single substrate. Two types of sensing elements of similar design and different widths (0.2 or 3 μm) were located in the chips. The drain-gate measurements of wires with a width of 3 μm are sufficiently reproducible for the entire chip to obtain measurement statistics in air and deionized water. The sufficient correlation of results between different sensors of the same chip were observed in measurements for 0.2 μm width wires in air.

For the pH range from 3 to 12, we found significant changes in source-drain characteristics, which reach the plateau at values of 7 and higher. High pH sensitivity (ca. 1500 and 970 mV/pH) was observed at 3 μm and 0.2 μm width sensors in the range of the pH values from 3 to 7. We found a higher "on" current to "off" current ratio for wide wires. At all studied pH values, $I_{on}/I_{off}$ were up to 4600 and 30,800 for 0.2 and 3 μm wires, respectively. In the scheme on the source-drain current measurements at fixed gate voltages, the highest sensitivity to the pH changes was reached at gate voltage 13 and 19 V for 0.2 μm and 3 μm sensors, respectively. In summary, the most suitable are nanobelt sensing elements of 3 μm in width for the reliable analysis of biomolecules in the clinical samples.

## 6. Patents

Patent RU2762360C1 E. V. Dmitrienko, A. V. Poryvaeva, A. A. Ruban, E. V. Smolina, D. V. Pyshnyi, A. A. Lomzov, "Method for automating parallel tagless detection of a biological marker and a device for its implementation", 20 December 2021.

**Author Contributions:** Data curation and investigation, A.B. (Anastasia Bulgakova), A.B. (Anton Berdyugin), A.L. and E.D.; conceptualization, E.D. and A.L.; writing—original paper, A. Bulgakova, A.L. and E.D.; writing—review and editing, A.C.; funding acquisition, A.L., A.C., E.D. and D.P.; project administration, A.L., A.C., E.D. and D.P; sensing element design, O.N. and B.F. All authors have read and agreed to the published version of the manuscript.

**Funding:** This study was funded by the Russian Science Foundation (grant No. 22-24-00996).

**Data Availability Statement:** Not applicable.

**Conflicts of Interest:** The authors declare no conflict of interest.

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
