# Peer review of "Solution pH Effect on Drain-Gate Characteristics of SOI FET Biosensor"

_electronics, doi:10.3390/electronics12030777_

Round 1
Reviewer 1 Report
In this work, a silicon-on-insulator biosensor device was developed and used to determine the effect of solution pH on drain-gate characteristics. However, the presentation and experimental design of this paper are deficient and the following changes need to be made.
1 In Figure 1, the schematic diagram of the measurement scheme shows the grounding schematic of the power supply, which is incorrectly represented.
2 The specific structure schematics for different size devices (3μm and 0.2μm devices) should be drawn so that the reader can easily understand the differences between the devices.
3 Figure 2 is missing labeled serial numbers A and B
4 In lines 141 and 142, the authors say “the current values for the different wires almost completely repeat each other until the gate voltage of 27 V”. But this conclusion is not clear from Figure 2
5 The authors should clarify the meaning of the “before” and “after” data in Figures 3 A and B
6 As can be seen from Figure 3A and B, the electrical properties of the device do not increase or decrease in one direction for pH values, and such a result seems impossible for practical applications. For example, in Figure 3A, the current values of pH 3 and pH 5 are almost equal for both conditions. How can we determine which specific pH value to use in a practical application?
Author Response
Thank you for the valuable suggestions and comments. We have carefully examined the comments and suggestions and revised the manuscript accordingly. We presented the word file with track changes. Please find as follows the responses to the comments. Please note that all the comments are bold-faced, and the authors' reply follows immediately below the comments.
1 In Figure 1, the schematic diagram of the measurement scheme shows the grounding schematic of the power supply, which is incorrectly represented.
Response: We have corrected the measurement scheme (Figure 1).
2 The specific structure schematics for different size devices (3μm and 0.2μm devices) should be drawn so that the reader can easily understand the differences between the devices.
Response: We have added optical microscope images of the nanobelts in Figure 1.
3 Figure 2 is missing labeled serial numbers A and B
Response: We have modified Figure 2 and added serial numbers.
4 In lines 141 and 142, the authors say “the current values for the different wires almost completely repeat each other until the gate voltage of 27 V”. But this conclusion is not clear from Figure 2
We have recalculated all toehold voltage values with the use of Ratio method [https://doi.org/10.1016/S0026-2714(02)00027-6] and corrected values in the text.
5 The authors should clarify the meaning of the “before” and “after” data in Figures 3 A and B
Response: We have modified Figure 3 and removed this information.
6 As can be seen from Figure 3A and B, the electrical properties of the device do not increase or decrease in one direction for pH values, and such a result seems impossible for practical applications. For example, in Figure 3A, the current values of pH 3 and pH 5 are almost equal for both conditions. How can we determine which specific pH value to use in a practical application?
Response: We have corrected Figure 3, the toehold voltage dependence on the pH, and the discussion in the text in the sections 3.3 and 4.
Reviewer 2 Report
The results are interesting and I recommend its publication with minor revision.
1. Please mention what is the effect of pH increasing and decreasing, means mechanism
2. What is the meaning of unmodified and nanobelt and nanowire sensors
3. What is error limit in figure 2
4. What is the sensitivity of the sensor
5. Did tested the sensor in different buffers
6. What is the purity of these analytical samples?
7. Check the experimental section and write clearly the used conditions (volumes in mL, pH, concentration etc.) used in performing the experiments. This section should be clearly usable by readers.
8. Authors should provide the calculation for the surface coverage area.
9. In the real sample analysis what is the dilution factor?
Author Response
Thank you for the valuable suggestions and comments. We have carefully examined the comments and suggestions and revised the manuscript accordingly. We presented the word file with track changes. Please find as follows the responses to the comments. Please note that all the comments are bold-faced, and the authors' reply follows immediately below the comments.
- Please mention what is the effect of pH increasing and decreasing, means mechanism
Response: We have added information about the effect of pH changes in the introduction and discussion sections.
- What is the meaning of unmodified and nanobelt and nanowire sensors
Response: The term “nanobelt” is more suitable for our sensors than the nanowire since the only thickness is in the nanometer range. We replaced the text “nanowire” with “nanobelt” concerning our sensors.
To make it clear, we have added information on nanobelt surface cleaning before measurements in the Materials and Methods section “2.3. Surface preparation of SOI-FET sensors”.
- What is error limit in figure 2
Response: We have added this information in the legend of Figure 2 and in the text of section 3.2
- What is the sensitivity of the sensor
Response: We have determined the pH sensitivity of the sensors and added this information in sections 3.3 and 4.
- Did tested the sensor in different buffers
Response: We have tested the only one buffer in the wide range of pH.
- What is the purity of these analytical samples?
Response: In the study, we used material of the highest available grade purchased from Sigma. Section 2.1 was corrected. The purity of all used components was added in the text.
- Check the experimental section and write clearly the used conditions (volumes in mL, pH, concentration etc.) used in performing the experiments. This section should be clearly usable by readers.
Response: We added the information in section 2
- Authors should provide the calculation for the surface coverage area.
Response: We added the information on the surface area in section 2.2
- In the real sample analysis what is the dilution factor?
Response: In the paper, we analyzed solutions with a different pH and the same ionic strength of the solution.
Reviewer 3 Report
Bulgakova et al. designed the silicon-on-insulator field-effect transistors and used as biosensor by measuring the different pH level. The manuscript is suggested to be accepted after the following issues are addressed.
1) Many spelling, grammatical, units and typo errors are present in this paper; the authors should double-check and revise them thoroughly.
2) There is no information about the surface morphology and thickness of the active channel. The authors used some SEM or AFM techniques to study the surface.
3) In Figure 2, the authors should also add the Ids-Vbg curves in the log scale to observe the leakage current and on/off ratio.
4) The authors did not provide any information about the electrodes, such as metal, thickness etc.
5) What’s about the sensitivity of the sensors?
6) The authors should also describe the selectivity of the biosensors.
7) The author should measure the device in a vacuum, use it as a reference and then measure in the Air and water
8) The authors should add a comparative table or figure to compare these outcomes with the previous study
9) Some related or latest studies should be discussed or cited, 10.1021/acsami.0c05114; 10.3390/s20236921
Author Response
Thank you for the valuable suggestions and comments. We have carefully examined the comments and suggestions and revised the manuscript accordingly. We presented the word file with track changes. Please find as follows the responses to the comments. Please note that all the comments are bold-faced, and the authors' reply follows immediately below the comments.
- Many spelling, grammatical, units and typo errors are present in this paper; the authors should double-check and revise them thoroughly.
Response: We checked our text and corrected errors.
- There is no information about the surface morphology and thickness of the active channel. The authors used some SEM or AFM techniques to study the surface.
Response: We added information on the thickness of the active channel in Section 2.2.
- In Figure 2, the authors should also add the Ids-Vbg curves in the log scale to observe the leakage current and on/off ratio.
Response: We modified Figure 3 by adding this information. The discussion of the observed characteristics is added in sections 3.3 and 4.
- The authors did not provide any information about the electrodes, such as metal, thickness etc.
Response: The information about the sensor construction was added in Section 2.3.
- What’s about the sensitivity of the sensors?
Response: We determine the pH sensitivity of the sensors and add this information in section 3.3 and in the discussion (section 4).
- The authors should also describe the selectivity of the biosensors.
Response: We corrected the text concerning the selectivity of the biosensors.
- The author should measure the device in a vacuum, use it as a reference and then measure in the Air and water
Response: Many thanks to the reviewer for the helpful suggestion. We analyzed the reproducibility of the measurements for the chips with multiple detecting sensors in the air and water solutions. Our study directed the development of biosensors for the real-time detection of biomarkers. The vacuum measurements are not suitable for this purpose.
- The authors should add a comparative table or figure to compare these outcomes with the previous study
Response: We added the comparative Table 1 for different sensor studies. The discussion is added in section 4.
- Some related or latest studies should be discussed or cited, 10.1021/acsami.0c05114; 10.3390/s20236921
Response: We add these references in the text and analyzed in the discussion section.
Reviewer 4 Report
Dear authors
The concept of this research is very simple and the data is very poor. Thus I cannot recommend it for publication. I think the authors should consider the follow problems before submitting it.
1. Drain-gate characteristics ( transfer curve) is only one basic feature of filed-effect transitors. The authors must consider other characteristics such as mobility, output curve, ON current/OFF current ratio, ....
2. I cannot find the novelty as well as significance of this research, compared with others in the litterature. Many studies did investigate the influence of pH on the perfomance of biosensors.
3. The changes in the drain-gate charactersitics of the biosensor is simply evaluated in water. It was not tested with biomolecule analytes. Thus, the results is not meaningful
Author Response
Thank you for the valuable suggestions and comments. We have carefully examined the comments and suggestions and revised the manuscript accordingly. We presented the word file with track changes. Please find as follows the responses to the comments. Please note that all the comments are bold-faced, and the authors' reply follows immediately below the comments.
- Drain-gate characteristics (transfer curve) is only one basic feature of filed-effect transitors. The authors must consider other characteristics such as mobility, output curve, the ON current/OFF current ratio, ....
Response: We add the information on the ON current/OFF current ratio and toehold voltage for the measurements in Figure 3 and in sections 3.3 and 4.
- I cannot find the novelty as well as significance of this research, compared with others in the litterature. Many studies did investigate the influence of pH on the perfomance of biosensors.
Response: We significantly modified our manuscript and added the comparison of our data with the previous studies.
- The changes in the drain-gate charactersitics of the biosensor is simply evaluated in water. It was not tested with biomolecule analytes. Thus, the results is not meaningful
Response: Chips with 12 same sensor nanobelt elements of different widths were studied in air, in deionized water, and in buffers over a wide range of pH values. The results obtained were discussed in the context of the applicability of the studied sensors for the detection of real samples (for example, nucleic acids).
Round 2
Reviewer 1 Report
The authors anwsered my questions. The manuscript can be published as this.
Reviewer 3 Report
Accepted in the present form.
Reviewer 4 Report
It can be accepted in the present version